# Seasonal Variation of Water Quality Modulated Redox Regulatory System in the Apple Snail *Pila globosa* and Its Use as a Bioindicator Species in Freshwater Ecosystems across India

Falguni Panda [1,2], Samar Gourav Pati [1,2], Taslima Nasim Anwar [1], Luna Samanta [2,*] and Biswaranjan Paital [1,*]

1  Redox Regulation Laboratory, Department of Zoology, College of Basic Science and Humanities, Odisha University of Agriculture and Technology, Bhubaneswar 751003, India
2  Redox Biology & Proteomics Laboratory, Department of Zoology, School of Life Sciences, Ravenshaw University, Cuttack 753003, India
*  Correspondence: lsamanta@ravenshawuniversity.ac.in (L.S.); biswaranjanpaital@gmail.com (B.P.); Tel.: +91-674-2397029 (B.P.); Fax: 91-0674-2397970 (B.P.)

**Abstract:** Studies on oxidative stress physiology on molluscs to monitor the aquatic environment, influenced by pollutants and stressors are very limited in general and in Indian apple snails, *P. globosa* in particular. The main focus of the present study was to establish the baseline data on a redox regulatory system in *P. globosa* sampled across the Indian subcontinent. Snails were sampled from different zones of India in three seasons (rainy, summer and winter) and the redox regulatory system (levels of antioxidant enzyme activities and small redox regulatory molecules) and oxidative stress indicator (lipid peroxidation) were analyzed. The observed elevated lipid peroxidation level in the summer season accompanied with elevated pH, salinity and temperature indicates induction of stress. In the summer season, the activities of superoxide dismutase (SOD), catalase (CAT) and glutathione reductase (GR) enzymes were augmented, whereas the level of the free -SH group and the activities of glutathione peroxidase (GPx) and glutathione-S-transferase (GST) were found to be increased. Similarly, correlation analysis between the antioxidant system and aquatic parameters revealed that SOD, CAT and GR were influenced by pH as well as salinity, whereas CAT was strongly correlated with temperature. Collectively, our data indicate the use of *P. globosa* as a model organism to monitor and access the freshwater environment by determining the redox regulatory status of this animal.

**Keywords:** anthropogenic activity; apple snail; biomarker of pollution; environmental toxicity; oxidative stress; *Pila globosa*; redox regulation; freshwater model; seasonal variation





## 1. Introduction

From the last few decades, various attempts have been made to establish a suitable biomarker in invertebrates using redox regulatory and oxidative stress indices, especially to monitor freshwater environments [1]. Being the key organism in freshwater as well as grassland ecosystems, the Indian apple snail *Pila globosa* is proposed as a potential model organism for the above purpose. When the molecular oxygen is incompletely reduced by mitochondria, it produces superoxide radicals ($O_2^-$) that act as the precursor for most of the other reactive oxygen species (ROS). The accumulation of ROS in cells leads to the oxidation of biomolecules including proteins, lipids and nucleic acids, resulting in the impairment of cellular functions. Such a condition is called oxidative stress (OS) [2]. Therefore, the cellular antioxidant defense system in organisms plays a vital role in keeping the quantity of ROS to the basal level by neutralization [3,4]. The severity of OS is dependent on the metabolic process. So, fluctuations in conditions such as food availability, the levels of pollutants, dissolved oxygen, salinity, pH, temperature etc. on a seasonal basis may influence OS levels in animals, especially in the ectotherms because the above factors are known to influence the metabolism of all the inhabitants in water bodies [5]. The above indices

are also influenced by anthropogenic activities; therefore, before using such (ectothermic) animals, including *P. globose*, as an indicator species for monitoring the environment, it is crucial to look into the in vivo seasonal changes in their redox regulation levels.

The cellular antioxidant system broadly comprises of two parts such as small redox regulatory non-enzymes including ascorbic acid (AA), reduced glutathione (GSH) and carotenoids and a cascade of enzymes consisting of superoxide dismutase (SOD), catalase (CAT), glutathione peroxidase (GPx) and glutathione reductase (GR) [6,7]. Since the antioxidant capacity in an animal is directly regulated by many antioxidant enzymes, fluctuations in their activities are used to monitor the environmental pollution that could be anthropogenic in nature in freshwater bodies [8]. Their seasonal variation is also crucial in animals, and the effects of different modulators on antioxidant enzymes are summarized in Table 1. Therefore, it is essential to know the baseline data on the seasonal variation of the above indices in freshwater bodies using an indicator species to assess environmental pollution.

**Table 1.** Oxidative physiology status of different molluscs in response to different modulators.

| Organism | Modulator | OS | AE | SA | BTE |
|---|---|---|---|---|---|
| *Mytilus edulis* (Blue mussel) | BaP(50 ppb) | LPx↑ (16.4%) | GPx↑ (7.3%) CAT↑ (21.5%) SOD↑ (9.2%) | GSH↑ (550%) | GST↓ (74.28%) |
| *Helix aspera* (Brown garden snail) | Microcystin (0.5 µg/g) | MDA↑ (50%) | GPx↑ (33%) CAT↑ (180%) SOD↑ (15%) | GSH↑ (550%) | GST↓ (74.28%) |
| *Mytilus edulis* (Blue mussel) | Menadione (1 ppm) | LPx↑ (42.8%) | GPx ↓ (16.2%) SOD↓ (23.8%) CAT↑ (8.9%) | NA | NA |
| *Biomphalaria alexandrina* (Ram's horn snail) | Oxyfluorfen (4.48 mg/L) | MDA↑ (82.2%) | SOD↑ (59.3%) CAT↑ (36.3%) | GSH↓ (56.3%) | NA |
| *Geukensia demissa* (Ribbed mussel) | Paraquet (1 mM for 6 h) | LPx↑ (22.3%) | SOD↑ (8.5%) CAT↑ (66.9%) | GSH↑ (4%) | NA |
| *Monacha cartusiana* (Cartusiana snail) | ZnO (740 µg/L) | MDA↑ (125.75%) | GPx↑ (187.5%) CAT↑ (163.2%) | NA | GST↑ (43.3%) |
| *Unio tumidus* (Swollen river mussel) | Cupper (30 µg/L) | NA | SOD↑ (6.4%) CAT↓ (9.1%) GPx↓ (15%) | NA | NA |
| *Theba pisana* (White garden snail) | Acrylamide (2.28 µg/g) | LPO↑ (31%) | CAT↑ (125%) | GSH↓ (10.2%) | GST↑ (68%) |
| *Unio tumidus* (Swollen river mussel) | Thiram (100 µg/L) | NA | SOD↓ (6%) CAT↓ (9.8) GPx↑ (10.7%) | NA | NA |
| *Chilina parchapii* | Pyrethroid cypermethrine (10 mg/L) | NA | GPx↑ (200%) CAT↓ (20%) | GSH↑ (20%) | GST↑ (140%) |

Notes: Oxidative stress along with antioxidant levels of different molluscs were modulated by different chemicals. OS—oxidative stress, AE—antioxidant enzyme, SA—small antioxidant molecule and BTE—bio transferring enzyme, MDA—malondialdehyde. NA indicates not applicable.

*P. globosa*, generally known as the Indian apple snail, is found generally in lentic water in oriental and Ethiopian countries. Although the apple snail population is abundant in India, only 42 research items are available in PubMed on it, which indicates the lacuna in understanding its environmental physiology in relation to the spatio-temporal changes in redox metabolic activities [9,10]. Very little data on the redox regulatory systems and OS indices of this organism are available so far. Therefore, the objective of this investigation was to evaluate the effects of seasons and space on the OS and the antioxidant systems in *P. globosa*, so that this organism can be used as an indicator species to monitor freshwater environmental pollution and anthropogenic effects on it.

## 2. Materials and Methods

### 2.1. Sampling Sites and Animals

Apple snails (*P. globosa*) were sampled from five different zones of India, i.e., Uttar Pradesh (UP, 25.357532° N 82.956795° E), Gujarat (22.798345° N 72.055040° E), Tamil Nadu (TN, 10.970114° N 76.905526° E), Odisha (19.678667° N 85.472083° E ) and Madhya Pradesh (MP, 23.249579° N 77.394557° E) as northern, western, southern, eastern and central zones, respectively, during the rainy, winter and summer seasons during the year 2018–2019. The water physicochemical parameters such as pH, salinity and temperature were measured using specific electrodes at the sampling sites. After collection, they were immediately transferred to a plastic opaque box containing ambient water and weeds and transported to nearby field laboratories. Mature snails (*n* = 5 for each season) were screened and sacrificed, keeping them in an ice box. Their foot muscle tissue was dissected out immediately, washed, blotted, packed (all steps are performed on ice) in cryovials (5 mL) and transferred into a liquid nitrogen tank. Then, the tissues were transferred to the laboratory and stored at −20 °C for further analysis.

### 2.2. Tissue Processing

A 10% tissue homogenate (*w/v*) was prepared in buffer containing 50 mM Tris–Cl, 1 mM EDTA, I mM DTT, 0.5 mM sucrose, 150 mM KCl and 1 mM PMSF at pH 7.8 [6] and centrifuged at $1000\times g$, 4 °C for 10 min to obtain the supernatant. It was referred to as the post-nuclear fraction (PNF). The obtained PNFs were again centrifuged at $10,000\times g$, 4 °C for 10 min to collect the clear supernatant as a post-mitochondrial fraction (PMF).

### 2.3. Determination of Oxidative Stress

Lipid peroxidation (LPx) was determined as the OS index. For the LPx assay, suitably diluted PNF samples with potassium chloride (1.15%; *w/v*) was used and thiobarbituric acid reactive substances (TBARS) were estimated by the method of Ohkawa et al. [11]. The level of TBARS was calculated from its molar extinction coefficient as $1.56 \times 10^5 \, M^{-1} \, cm^{-1}$ and expressed as the nmol of TBARS formed per mg protein.

### 2.4. Determination of Antioxidant Activities

The activities of SOD, CAT, GPx and GR were determined as markers of antioxidants as mentioned in Paital and Chainy [6].

#### 2.4.1. Superoxide Dismutase (SOD)

The SOD activity was determined according to Das et al. [12]. Briefly, a cocktail was prepared containing L-methionine (20 mM), Triton-X-100 (1%), Hydroxylamine hydrochloride (10 mM), EDTA (100 mM) and riboflavin (50 µM) in 100 mM Tris buffer, pH 8.0. After adding a sample to the cocktail, it was transferred to a wooden box fitted with two parallel 20 W fluorescent lamps and exposed for 10 min at room temperature for generation of superoxide redical by photo-oxidation of riboflavin. The generated superoxide was converted to nitrite by interacting with hydroxyl amine hydroclodied and measured by Griess reagent that was measured at 543 nm and expressed as a unit (U) per mg protein, where one unit defines the amount of protein that creates a 50% inhibition of the production of nitrite compounds.

#### 2.4.2. Catalase (CAT)

Catalase activity was estimated according to Aebi [13], in which samples were treated with Triton-X-100 (1%) and ethanol (1%) and incubated on ice for 30 min prior to the assay. After incubation, the samples were added to the cuvette containing freshly prepared $H_2O_2$ (25 mM). The catalase activity was calculated from the decrease in the absorbance of $H_2O_2$ at 240 nm, and by taking its extinction coefficient ($43.6 \, M^{-1} \, cm^{-1}$), the catalase activity was calculated. The activity was expressed as nCAT/mg protein, where one cat defines one mole of $H_2O_2$ consumed per second per mg protein.

### 2.4.3. Glutathione Peroxidase (GPx)

Glutathione peroxidase activity was estimated according to the coupled enzyme assay of Paglia and Valentine [14] by monitoring the utilization of NADPH at 340 nm. Briefly, a cocktail was prepared containing GSH (30 mM), KCN (4.5 mM) and $NaN_3$ (30 mM) in 23:1:1:1, respectively, and to this, NADPH (4.5 mM), GR (10 U/mL) and the sample were added. The substrate Cumene hydroperoxide (7.5 mM) was added to start the reaction. The GPx enzyme activity was calculated from the extinction coefficient of NADPH i.e., $6.22 \times 10^3$ $mM^{-1}$ $cm^{-1}$ and expressed as the nmol of NADPH oxidized per minute per mg protein.

### 2.4.4. Glutathione Reductase (GR)

Glutathione reductase activity was measured according to Massey and Williams [15] by monitoring NADPH oxidation at 340 nm which depends upon the rate of the conversion of oxidized glutathione into reduced glutathione. Briefly, oxidized glutathione (120 mM) and NADPH (4.5 mM) were added to the cuvette containing phosphate buffer (50 mM, pH-7.6) and the reaction wast stred by adding the sample. The enzymatic activity was calculated from the extinction coefficient of NADPH ($6.22 \times 10^3$ $mM^{-1}$ $cm^{-1}$) and expressed as the nmol of NADPH oxidized per minute per mg protein.

### 2.5. Measurement of Small Redox Regulatory Molecule and Bio-Transferring Enzyme Activity

### 2.5.1. Ascorbic Acid (AA)

Ascorbic acid content was determined according to Mitsui and Ohta [16] with a little modification. Prior to the assay, the PMF samples were treated with ice-cold TCA and centrifuged at $10,000\times g$, 4 °C for 10 min to collect a clear supernatant. Briefly, 1 mL of assay system contained 0.5 mL of sample, 0.2 mL of sodium molybdate (0.66%), 0.2 mL $H_2SO_4$ (0.05 N) and 0.1 mL sodium phosphate (0.025 mM) was kept at 60 °C for 40 min. The absorbance was taken at 660 nm after cooling the assay system, followed by centrifugation. The ascorbic content was estimated from the standard curve of it and expressed in ng per mg protein.

### 2.5.2. Non-Protein Sulfhydryl (-SH) Group

The non-protein –SH group content was determined by Sedlak and Lindsay [17]. Briefly, 2.85 mL of phosphate buffer was pipetted into a test tube, and 0.5 mL each of DTNB (0.01 M), GSH (0.25 mM) and the sample were added to it at 1 min intervals. Then, the tubes were incubated at room temperature for 10 min and the color product was determined at 412 nm. The values of the –SH content of the samples were calculated from the standard curve of GSH ranging from 0.1 to 0.4 M. The non-protein -SH content was expressed as nmol of –SH per mg protein.

### 2.5.3. Glutathione-S-Transferase (GST)

The activity of glutathione-S-transferase was estimated according to Keen et al. [18]. Briefly, a 3 mL assay system contained 0.1 mL each of GSH (30 mM), CDNB (15 mM) and the sample in phosphate buffer (100 mM, pH-6.5). An increase in the absorbance of GSH-CDNB conjugate formation was monitored at 340 nm, and the result was calculated from the extinction coefficient of CDNB ($0.00503$ $\mu M^{-1}$), and the results were expressed as micromoles of CDNB conjugates formed/min/mg protein, where one unit defines as a $\mu$mol of CDNB conjugated with reduced glutathione per mg protein.

### 2.6. Determination of Total Antioxidant Activity

The total antioxidant activity was determined by the method of Bal et al. [19] by monitoring the DPPH scavenging activity. Briefly, to the 1.4 mL of methanol, 100 $\mu$L of DPPH (0.5 mg/mL of methanol) solution was added, followed by 100 $\mu$L of the PNF sample. The absorbance of the reaction mixture was monitored at 515 nm, and the results were expressed as the % of decrease in the absorbance by 100 $\mu$L of PNF from the control

set. The calculation was based on the formula as % of inhibition = (Abs Control − Abs of sample)/Abs control × 100 [20].

### 2.7. Statistical Analysis

Each set of oxidative stress physiology was expressed as mean ± SEM, *n* = 5. The data were tested for normal distribution. The means of biochemical data were compared and analyzed by two-way ANOVA analysis, followed by Duncan's new multiple range test. Differences among means were considered significant at the $p \le 0.05$ level. Correlation coefficients ('r') between the OS parameters and water parameters of the sampling sites were determined at a 5 percent significance level using the Microsoft Excel program. Discriminant function analysis (DFA) was performed to evaluate the contribution of the variables of antioxidant enzymes and small antioxidant molecules on the groups. Mean values of the biochemical parameters between the groups were compared to calculate the absolute change in percentage. The data with different superscripts are statistically different from each other at the $p \le 0.05$ level.

### 3. Result

The water parameters of the concerned habitats during different seasons were summarized in Table 2 and were correlated with OS and antioxidant parameters.

**Table 2.** Spatio-temporal variation of water quality parameters at different sampling sites.

| Location | Season | PH | Salinity (PPT) | Temperature (°C) |
|---|---|---|---|---|
| | Rainy | 8.54 [a] | 2.50 [a] | 26 [b] |
| GUJARAT | Winter | 8.75 [b] | 2.70 [b] | 17 [a] |
| | Summer | 8.92 [c] | 3.00 [c] | 36 [c] |
| | Rainy | 8.12 [a] | 2.50 [a] | 25 [b] |
| MP | Winter | 8.25 [b] | 2.60 [b] | 21 [a] |
| | Summer | 8.92 [c] | 2.90 [c] | 42 [c] |
| | Rainy | 8.31 [a] | 2.20 [a] | 32 [b] |
| TN | Winter | 8.56 [b] | 2.50 [b] | 25 [a] |
| | Summer | 8.72 [c] | 2.90 [c] | 39 [c] |
| | Rainy | 8.10 [a] | 2.10 [a] | 36 [b] |
| UP | Winter | 8.63 [b] | 2.40 [b] | 25 [a] |
| | Summer | 8.81 [c] | 3.00 [c] | 44 [c] |
| | Rainy | 7.26 [a] | 1.60 [a] | 28 [b] |
| ODISHA | Winter | 7.96 [b] | 1.90 [b] | 21 [a] |
| | Summer | 8.23 [c] | 2.40 [c] | 41 [c] |

Notes: Different parameters of water from the concerned habitats were determined during different seasons. Different superscripts indicate the statistical difference among the mean values at the $p < 0.05$ level.

The salinity level ranged from 1.6 to 3 ppt, while the pH ranged from 7.26 to 8.92. Similarly, the temperature of the sampling site ranged from 17 to 42 °C (Table 2).

### 3.1. Oxidative Stress Indicator

Oxidative stress was determined from the LPx level. The lipid peroxidation level was found to be the highest during the summer season for snails collected from all the locations (Figure 1a). There was a 0.5 to 8 times lower LPx value observed during the rainy season as compared to the summer season. The lowest LPx level was observed in the foot muscle of snails collected from UP and TN during the rainy season as compared to the other sampling sites. The total antioxidant capacity was detected to be the highest during the summer season and was the highest at the Gujurat sampling site as compared to the other sampling sites. The trend of the variation of this parameter was summer > winter > rainy season (Figure 1b). The DPPH scavenging capacity was found to be lowest during the rainy season, being 64% lower than in the summer season (Figure 1b). Out of all the states,

snails collected from Gujarat and TN showed the highest and lowest total antioxidant activities, respectively.

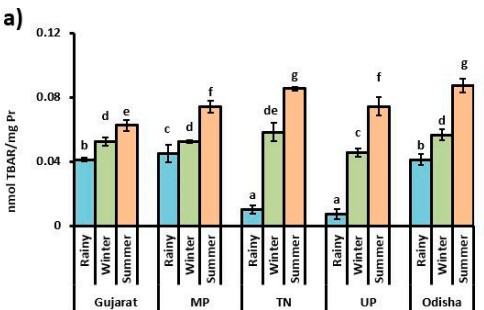 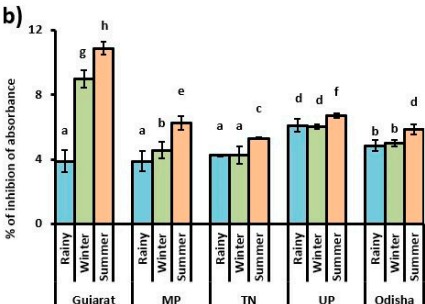

**Figure 1.** Seasonal variation of oxidative stress and total antioxidant activity of *P. globosa*. The oxidative stress condition was determined by assay of the lipid peroxidation level (**a**) in the snails while the total antioxidant capacity in the form of DPPH scavenging activity (**b**) was measured by DPPH scavenging activity. The data represent mean ± SEM with superscripts indicating statistical differences between mean values at *p* < 0.05 level.

### 3.2. Antioxidant Enzyme Activities

The lowest SOD activity was observed during the rainy season, while its activity was the highest during the summer season as compared to the other two seasons (Figure 2a). The SOD activity was 300–400% elevated during summer than during the rainy season. Spatially, the highest and lowest SOD activity was observed in snails collected from TN and MP, respectively, as compared to the other sampling sites (Figure 2a). A surge in CAT activity was recorded during the summer season as compared to the other two seasons. The highest CAT activity was found during the summer season, having 25–110% more activity as compared to the other two seasons. Similarly, out of all the locations, the highest CAT activity was observed in snails sampled from UP while the lowest was found in the Odisha sampling site as compared to the other two sites (Figure 2b). The higher GPx activity was observed during the winter season as compared to the other seasons in snails. The lowest GPx activity was found during the rainy season, having 64 to 73% less activity in comparison to the other two seasons (Figure 2c). The highest and lowest activity of this antioxidant enzyme was observed in snails sampled from the UP and Gujarat regions, respectively. The highest GR activity was observed during the summer season while the lowest activity was found in the rainy season (57% less activity) in snails as compared to the other sampling sites (Figure 2d). Spatially, snails sampled from UP were observed to have the highest GR activity while those of MP had the lowest activity as compared to the other sampling sites. The total antioxidant activity was found to be elevated during the summer season in comparison to the other two seasons.

### 3.3. Content of Small Antioxidant Molecules and Bio-Transferring Enzyme Activity

The activity of the GST enzyme was observed to be elevated during the winter season, with a 233% augmentation than in the rainy season (Figure 3a). A moderate level of GST activity was found during the summer season as compared to the other two seasons. Spatially, the highest and lowest activity of this antioxidant enzyme was observed in the snails sampled from the UP and Odisha states, respectively (Figure 3a). The -SH group was 75 to 83.33% decreased during the rainy season compared to the winter season (Figure 3b). The non-protein -SH level was found to be the highest in snails collected from TN while that of MP had the lowest concentration as compared to the other sampling sites. Similarly, the AA content was observed to be the lowest during the summer season as compared to the other seasons (Figure 3c). The AA level was 8.5 to 10.5% alleviated during the summer season compared to the other two seasons. The highest AA content was recorded in snails sampled from the Gujarat region as compared to all the other locations.

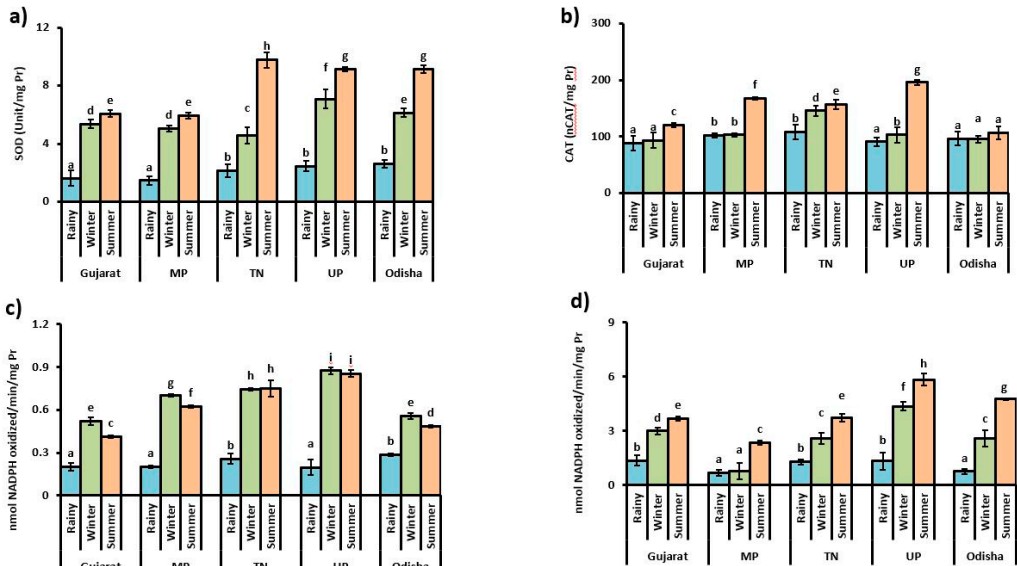

**Figure 2.** Seasonal variation of antioxidant enzyme activities of *P. globosa*. (**a**) superoxide dismutase, (**b**) catalase, (**c**) glutathione peroxidase, (**d**) glutathione reductase. The data represent mean ± SEM with superscripts indicating statistical differences between mean values at $p < 0.05$ level.

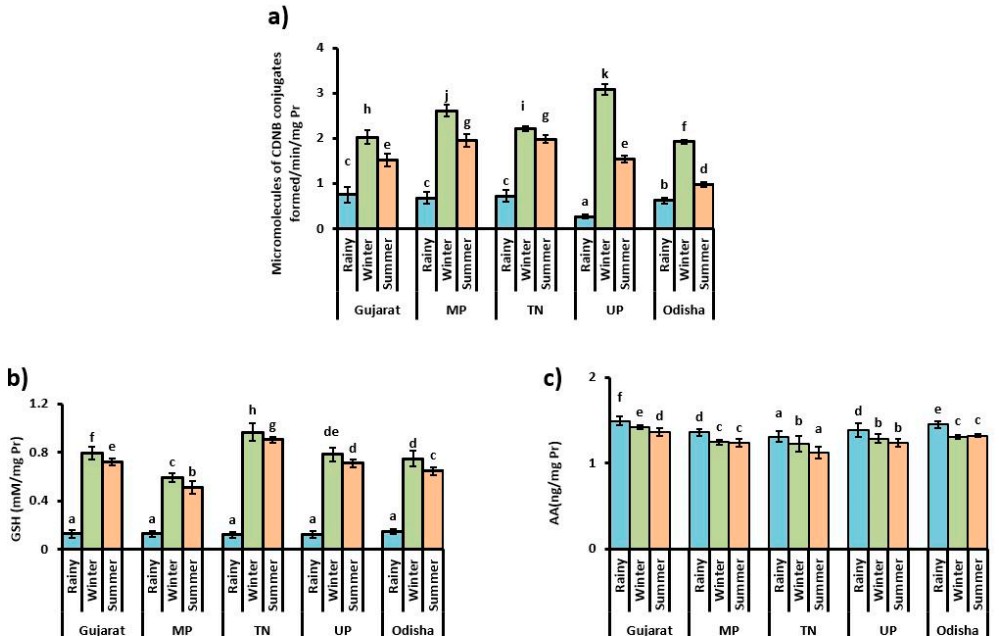

**Figure 3.** Seasonal variation in bio-transferring enzyme activity and content of small redox regulatory molecule in *P. globosa*. (**a**) glutathione-S-transferase, (**b**) non-protein-SH, (**c**) ascorbic acid. The data represent mean ± SEM with superscripts indicating statistical differences between mean values at the $p < 0.05$ level.

### 3.4. Relationships between Antioxidant Parameters

The generation of $H_2O_2$ and the rate of its neutralization by CAT and GPx enzymes was determined by the SOD/CAT+GPx value. The much lower value of SOD/CAT+GPx was observed during the rainy season in comparison to the other two seasons (Figure 4a). SOD+CAT+GPx value was found to be higher during summer than the other two seasons (Figure 4b). Higher SOD+CAT+GPx+GR+GST was detected during the summer season (Figure 4c). The AA+GSH value was also recorded as the lowest during the rainy season while insignificant variation was observed between the winter and summer seasons (Figure 4d).

The SOD+CAT+GPx+GR+GST/AA+GSH value was found to be lowest during the winter season (Figure 5a). Similarly, the highest and lowest SOD+CAT+GPx+GR+GST+AA+GSH values were detected during the summer and rainy seasons, respectively (Figure 5b).

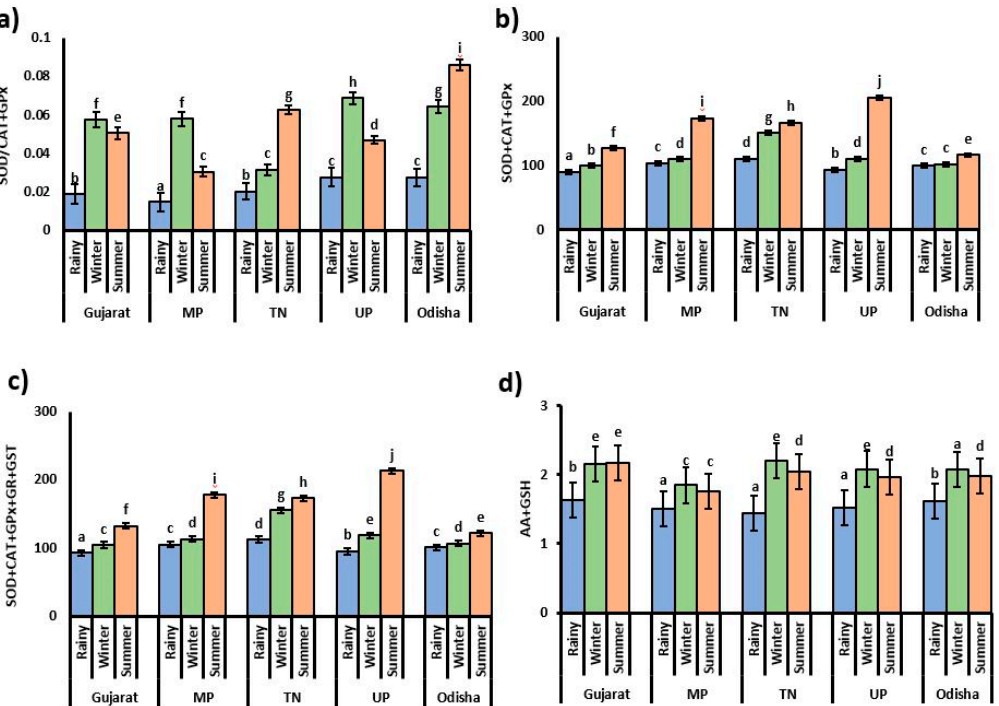

**Figure 4.** Relationship between different components of antioxidant defense system and season (**a–d**). The data represent mean $\pm$ SEM with superscripts indicating statistical differences between mean values at the $p < 0.05$ level.

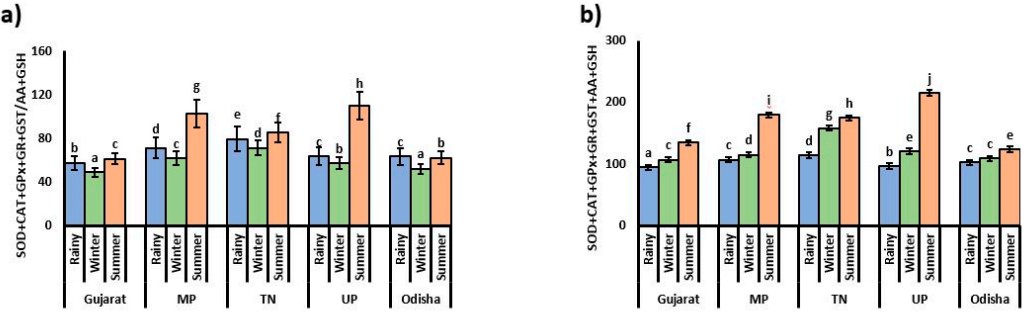

**Figure 5.** Seasonal variation of all measured antioxidant levels of *P. globosa*. (**a**) Correlation between enzymatic:non enzymatic antioxidants and (**b**) total enzymatic and non-enzymatic antioxidants with season and space. The data represent mean $\pm$ SEM with superscripts indicating statistical differences between mean values at the $p < 0.05$ level.

### 3.5. Correlation and DFA Analysis

The SOD, CAT and GR enzyme activities were found to have strong and positive correlation with pH as well as salinity (Table 3). The non-protein –SH group was found to be positively correlated with pH and salinity. A positive correlation was found between GST and pH only, while CAT activity was observed to be positively correlated with temperature. The AA content was insignificant with all the water parameters. The variables that were vital in determining the oxidative status of *P. globosa* were analyzed by DFA (Table 4). A distinct separation of seasonal groups in antioxidant enzymes was detected (Figure 6a). However, the variable for small antioxidant molecules was distinct only during the rainy season (Figure 6b). The radar chart of all the parameters explains the greater impact of season on SOD and GR enzymes (Figure 7).

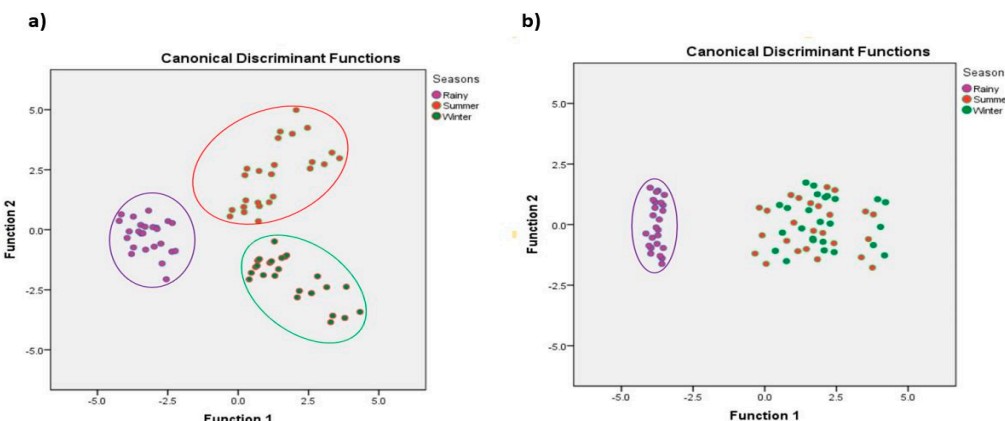

**Figure 6.** Discriminant function analysis for the antioxidant molecules of *P. globosa*. Discriminant function analysis for (**a**) antioxidant enzymes and (**b**) small antioxidant molecules were determined, which were divided into three groups among the rainy, winter and summer seasons.

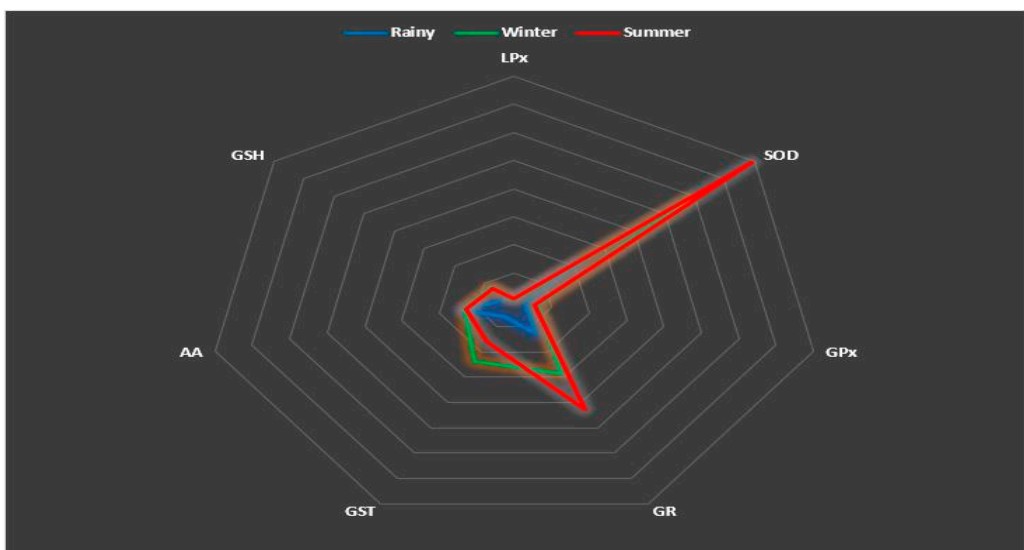

**Figure 7.** Radar chart of the seasonal variation of the antioxidant levels of *P. globosa*. Radar chart of antioxidant molecule levels in different seasons suggests the distinct impact of season on the SOD and GR levels.

**Table 3.** Correlation coefficients (r) values with respect to environmental parameters of different sampling sites.

| | Parameters pH | | | | | Salinity | | | | | Temperature | | | | |
|---|---|---|---|---|---|---|---|---|---|---|---|---|---|---|---|
| | Gujarat | MP | TN | UP | Odisha | Gujarat | MP | TN | UP | Odisha | Gujarat | MP | TN | UP | Odisha |
| LPx | 0.61 | 0.73 | 0.94 | 0.85 | 0.78 | 0.6 | 0.73 | 0.92 | 0.9 | 0.86 | ns | 0.67 | ns | ns | 0.66 |
| SOD | 0.93 | ns | 0.94 | 0.98 | 0.96 | 0.86 | 0.61 | 0.98 | 0.91 | 0.97 | ns | ns | 0.66 | ns | 0.60 |
| CAT | 0.89 | 0.99 | 0.98 | 0.77 | 0.97 | 95 | 0.95 | 0.92 | 0.97 | 0.78 | 0.79 | 0.98 | ns | 0.75 | 0.86 |
| GPx | 0.72 | ns | 0.76 | 0.89 | 0.83 | 0.59 | 0.49 | 0.63 | 0.61 | 0.57 | ns | ns | ns | ns | ns |
| GR | 0.81 | 0.74 | 0.82 | 0.94 | 0.88 | 0.78 | 0.72 | 0.82 | 0.87 | 0.92 | ns | 0.72 | ns | ns | 0.63 |
| GST | 0.62 | ns | 0.84 | 0.67 | ns | ns | ns | 0.71 | ns | ns | ns | ns | ns | −0.61 | −0.55 |
| AA | ns | ns | ns | ns | ns | ns | ns | ns | ns | ns | ns | ns | ns | ns | ns |
| GSH | 0.93 | ns | 0.89 | 0.93 | 0.9 | 0.85 | 0.63 | 0.78 | 0.68 | 0.67 | ns | ns | ns | ns | ns |
| DPPH | 0.86 | 0.5 | ns | ns | ns | 0.81 | 0.51 | ns | ns | ns | ns | ns | ns | ns | ns |

Notes: Different correlation coefficients (r) were considered at a 5% significant level. 'ns' is designated for non-significant values. The negative sign shows the negative correlation towards the respective parameter.

**Table 4.** Standardized Canonical Discriminant Function Coefficients with respect to different antioxidant parameters.

| Parameters | Function 1 | Function 2 |
|:---:|:---:|:---:|
| SOD | 0.655 | 0.767 |
| CAT | 0.001 | 0.93 |
| GPx | 0.131 | −0.488 |
| GR | −0.119 | −0.111 |
| GST | 0.658 | −0.614 |
| AA | −0.297 | 0.975 |
| GSH | 1.015 | 0.101 |

## 4. Discussion

The seasonal variation of environmental conditions impacts physico-chemical characters such as the pH, salinity and temperature of freshwater habitats [21,22]. Massive rainfall during the rainy season causes a depletion of the salinity and pH levels. The variation of environmental factors at different seasons fluctuates the homeostasis between antioxidant systems and ROS [6]. Many studies have been conducted on seasonal variations of antioxidant systems in higher animals. However, reports on seasonal variation of antioxidant system in molluscs are rare and on *P. globosa* is almost absent. In the land snail, *Helix aspersa*, the elevation of antioxidant enzyme activity such as GST and CAT was observed during the summer season while augmentation of -SH group content was noticed during the winter [23]. Viarengo et al. [24] also found increased CAT activity during the summer season in the digestive gland of the marine mussel, *Mytilus edulis*. There has been a report on the elevation of GPx and GST activity during the winter season in the gill and digestive glands of the blue mussel [25]. The thiol group content was also found to be elevated during the summer season in the digestive gland of the common mussel, *M. edulis*. Similarly in the brown mussel, *Perna perna*, the increment of the LPx level was observed during the summer season rather than in other seasons due to a higher oxygen consumption rate [26].

Previously, works on the American apple snail, *Pomacea canaliculata*, revealed the increased accumulation of LPx bioproducts during the summer season but recovered to a normal level soon after [27]. The higher LPx value observed during the summer season in our result suggests the OS condition of animals during this period, probably due to temperature and its induced water deprivation states [28]. Similarly, the depressed metabolic rate during aestivation could hamper the detoxification process and thus could increase the ROS level. Higher temperature during the summer season results in the increment of salinity and pH that were probably responsible for OS accumulation during the summer season [6]. On the other hand, the LPx level was found to be lower in the rainy season, during which the low salinity and pH levels were found. This result clearly shows the impact of environmental factors on OS physiology. Since the fat content is found to be very high during the summer season in aquatic animals [29], a very high chance for its oxidation by ROS is possible; the same fact is also reflected in the present study by formation of the highest TBARS during the summer season. The diminished TBARS level during the rainy season suggests the rainy season is an ideal environment for *P. globosa*.

In our results, a clear seasonal variation of the antioxidant defense system of *P. globosa* was observed. The antioxidant activities were found to be considerably lower during the rainy season in the snails when they often become very active and there is food abundance in their habitat. The increased activities of antioxidant enzymes during the winter and summer seasons were probably due to the occurrence of a dormant period such as hibernation and aestivation or there might be upregulated antioxidant enzymes as an adaptation to survive under high pH, salinity and altered ambient temperature in the environment [23]. The observed increase in the TBARS level during the above two seasons could be another cause for the elevation of antioxidant systems to compensate for the effect of ROS-induced OS [30]. Although the antioxidant level was augmented during the

summer and winter seasons, the regulation of the antioxidant system was varied in those seasons. During the summer season, the activities of SOD, CAT and GR were upregulated, while, the GPx and GST activities were elevated during both winter and summer season.

The SOD reduces superoxide ions to $H_2O_2$ which is further reduced to water molecules via CAT [31]. Thus, the activities of the above two enzymes were upregulated during the summer season to neutralize the $H_2O_2$ but appeared to be insufficient to combat the OS as evidenced by high TBARS concentration. Pakay et al. [32] found a reduction in the synthesis of protein during aestivation in *H. aspersa*. Therefore, diminished activities of SOD, CAT and GR enzymes during the winter season versus summer might be due to the negative effect of hypometabolism in *P. globosa*. Since the synthesis of protein is an energetically costlier process [33], proteins only required during hibernation are likely to be translated during the winter season. There is no such information about the post-translational regulation of enzymes such as SOD, CAT and GR. Therefore, it can be speculated that the above three enzymes might have altered their biosynthesis, or their rate of proteasome-regulated degradation might have increased during the winter season versus that of the summer. However, the elevated activity of GPx during the winter season might be to compensate for the OS. The higher CAT activity observed in foot muscles during the summer season in comparison to the winter season also suggests the inadequate efforts of this enzyme to reduce $H_2O_2$ level unaccompanied with GPx [34]. The elevation of the level of free sulfhydryl group during the winter season could be another cause for the augmentation of GPx activity during that season because the -SH group is used as a substrate for the activity of GPx [35]. Glutathione-S- transferase is a type II bio-transferring enzyme whose activity was found higher during the winter season in comparison to that of the summer season. The elevation of GST activity suggests the active involvement of this enzyme in detoxification by the conjugation of reduced glutathione to hydrophobic and electrophilic molecules including many products of oxidative metabolism in the snail during the winter season [36]. Ascorbic acid is synthesized with the help of the enzyme L-gulonolactone oxidase [37]. This enzyme is absent in most animals, so they need to consume AA through diet [38]. There is also no evidence about the presence of L-gulonolactone oxidase in snails. Thus, the decline in the AA level during the summer season could be due to its utilization in the neutralization of ROS [7]. Because the rainy season is their active period and they actively feed during this season, the elevation of AA during the rainy season could be acquired via food. However, the non-protein -SH group level was found to be elevated during the winter season, which suggests that GSH may be used as one of the important small antioxidant molecules to combat the OS state. The higher -SH level also could be responsible for the elevation of GPx activity as the GSH group is used as a substrate for GPx activity [35].

Total antioxidant capacity was observed to be the highest during the summer season while the lowest was during the rainy season. This might be due to differences in metabolic activity during the summer season under higher pH, salinity and temperature [3,4,6]. Since the activities of SOD, CAT and GR were also found to be higher during the summer season, it might be concluded that the elevation of the activities of the above three enzymes has an important role in neutralizing $H_2O_2$ and other ROS to compensate for their effects to induce OS under the increase in pH, salinity and temperature. It could indicate an adaptive response of the snails to cope with environmental stress conditions.

The higher value of most of the relationship parameters observed during the winter and summer seasons suggests the upregulation of antioxidant systems to compensate for the effect of ROS-induced OS. The augmented SOD/CAT+GPx value during summer and winter seasons rather than the rainy season implies higher production of $H_2O_2$ than neutralization of this molecule by CAT and GPx enzymes [39]. The highest upregulation of antioxidant enzyme activities (SOD+CAT+GPx+GR+GST) was observed during the summer season versus the other two seasons, which might be due to combatting excess ROS produced during this season as found in the TBAR values [40]. However, the levels of small antioxidant molecules was detected to be highest during the winter season as

compared to the other seasons. Therefore, it can be implied that the antioxidant enzymes and small redox regulatory molecules are responsible for reducing the ROS-induced OS during the summer and winter seasons, respectively, in this organism.

The strong positive correlations of SOD, CAT and GR with pH and salinity suggest that the above three enzymes might be responsible to compensate for the effect of OS during higher pH and salinity. However, the strong correlation of GST with only pH implies the sensitivity of the GST enzyme of *P. globosa* only with pH rather than salinity and temperature. Similarly, the -SH group was observed to be upregulated in response to higher pH and salinity, which might be to neutralize the ROS. None of the enzyme activities as well as small antioxidant molecules were correlated significantly except CAT, which implies its vital role in the neutralization of $H_2O_2$ in temperature-induced OS. A distinct group separation in antioxidant enzyme activity during different seasons implies the role of the environment on the physiology of *P. globosa*. However, there was only group separation of small antioxidant molecules during the rainy season, suggesting no variation of small redox molecules during the winter and summer seasons. Similarly, out of the enzymatic antioxidant system, only SOD and GR were found to be affected by the season. The discriminant function analysis indicates that the spatio-temporal variations of the studied OS parameters under the facultative water parameter in snails are distinct. The DFA and radar charts indicate that SOD and GPx are two of the most contributing factors for the seasonal separation of the dataset in snails.

## 5. Conclusions

Collectively, our data imply a clear seasonal variation of markers of the antioxidant system and OS in snails. The antioxidant molecules remained low during the rainy season while they were upregulated during the winter and summer seasons under higher pH, salinity and temperature in the later season. Moreover, the antioxidant enzymes were found to be upregulated during the summer season to compensate for the ROS-induced OS, whereas small antioxidant molecules with GPx were observed to be upregulated to combat the above effect during the winter season. The physiology of freshwater organisms alters the expression of antioxidant genes since transcription and translation of antioxidant enzymes are very sensitive towards the alteration of physiological homeostasis. Thus, it can be concluded that seasonal alteration of the biochemical level of antioxidant molecules occurring may be due to changes in the gene expression of those enzymes as an adaptive response in freshwater snails. Thus, *P. globosa* can be used as an excellent indicator species to monitor the freshwater environment as its redox regulatory system is considerably influenced by the season and space. Thus any anthropogenic pollution in freshwater can also be monitored using the redox regulatory system as markers in the species.

**Author Contributions:** Conceptualization, Formal analysis, Data curation, Funding acquisition, Investigation, Methodology, Project administration, Resources, Software, Supervision, Validation, Visualization, Writing—original draft, Writing—review and editing B.P., Formal analysis, Data curation, Writing—original draft, Writing—review and editing, F.P., S.G.P. and T.N.A., Writing—original draft, Writing—review and editing, supervision, L.S. All authors have read and agreed to the published version of the manuscript.

**Funding:** The work was generously supported by the funding to B.P. from the Science and Engineering Research Board, Department of Science and Technology, Govt. of India New Delhi, India (No. ECR/2016/001984) and the Department of Science and Technology, Government of Odisha (Grant letter number 1188/ST, Bhubaneswar, dated 1 March 2017, ST-(Bio)-02/2017).

**Institutional Review Board Statement:** Not applicable.

**Informed Consent Statement:** Not applicable.

**Data Availability Statement:** All data generated or analyzed during this study are included in this published article.

**Acknowledgments:** The authors duly acknowledge the use of the Central Instrumentation Facility of Odisha University of Agriculture and Technology, especially, from Sashikanta Dash for providing the instrumentation facility to measure the activity of enzymes such as SOD and CAT.

**Conflicts of Interest:** The authors declare no conflict of interest.

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
