# Peer review of "Seasonal Variation of Water Quality Modulated Redox Regulatory System in the Apple Snail Pila globosa and Its Use as a Bioindicator Species in Freshwater Ecosystems across India"

_water, doi:10.3390/w14203275_

Round 1

Reviewer 1 Report

The manuscript ie nearly perfectly preapeared. 

The only minor technical suggestion from my side is that the latin name for species should be written sometimes in full according to Journals requirement.

Author Response

Comments: The only minor technical suggestion from my side is that the latin name for species should be written sometimes in full according to Journals requirement.

Reply: The latin name while their first description are written in full and subsequently are written with the generic name abbreviated. For example, Pila globosa is written in full and then was abbreviated as P. globosa

Reviewer 2 Report

There acronymous for enzymes that are not defined or are shown to late; MDA is not defined and LPx defined in page 5, but they are used in table 1, page 2. Also the non-protein -SH group is just identified in the line 332 after been used along 8 pages!

The first paragraph of discussion is just a state of the art that could be done in the introduction. A concluding paragraph in the introduction explaining the structure of the paper also could be helpful.

Is not clear for me to state  the oxidative stress were expressed as mean +/- SEM (standard error of the mean) but than all figures use mean +/- SD (standard deviation) that is not the same and is a factor of confusion.

Is not clear the information you what to transmit with the superscripts on the figures, and reader have no way to give a significance to indices a to h.

Author Response

There acronymous for enzymes that are not defined or are shown to late; MDA is not defined and LPx defined in page 5, but they are used in table 1, page 2. Also the non-protein -SH group is just identified in the line 332 after been used along 8 pages!

Reply: MDA is defined now as malondialdehyde, and -SH is defined in the revised ms.

The first paragraph of discussion is just a state of the art that could be done in the introduction. A concluding paragraph in the introduction explaining the structure of the paper also could be helpful.

Reply: We agree that the first paragraph is a state of art, but to introduce the topic, we followed some good publications and made it like this. Already such info are presented in intro part, so we kept it as such.

Is not clear for me to state  the oxidative stress were expressed as mean +/- SEM (standard error of the mean) but than all figures use mean +/- SD (standard deviation) that is not the same and is a factor of confusion.

Reply: It is SEM, we are sorry for the typo error, we have corrected them as SEM in the entire MS.

Is not clear the information you what to transmit with the superscripts on the figures, and reader have no way to give a significance to indices a to h.

Reply: The superscripts are the indicators of the difference among the mean at p<0.05 level. The same has been indicated in the revised ms. 

Reviewer 3 Report

This manuscript examines seasonal variations in various enzyme levels related to the oxidative stress physiology system of the Indian Apple Snail Pila globosa as a potential bioindicator of stress conditions in freshwater habitats. Snails were sampled during three seasons (rainy, summer, and winter) at various locations within India and components of the antioxidant system (antioxidant enzyme activity and small regulatory molecules) and oxidative stress marker (lipid peroxidation) were analyzed and comparisons by with season and site location were made. Sampling protocols, analytical protocols, and statistical procedures are well-described in the manuscript, and support the data interpretation and results reported of seasonal variations in the anti-oxidant system and oxidative stress, especially  during periods of higher pH, salinity, and temperature. These data also justify the use of P. globosa as a biomonitor of environmental conditions.

Overall, the manuscript, tables, and figures are clear and well-organized. The manuscript needs to be carefully edited, as there are some lengthy and awkward sentences and unclear wording in some sections. The results are very interesting and the manuscript should be accepted for publication.

Author Response

This manuscript examines seasonal variations in various enzyme levels related to the oxidative stress physiology system of the Indian Apple Snail Pila globosa as a potential bioindicator of stress conditions in freshwater habitats. Snails were sampled during three seasons (rainy, summer, and winter) at various locations within India and components of the antioxidant system (antioxidant enzyme activity and small regulatory molecules) and oxidative stress marker (lipid peroxidation) were analyzed and comparisons by with season and site location were made. Sampling protocols, analytical protocols, and statistical procedures are well-described in the manuscript, and support the data interpretation and results reported of seasonal variations in the anti-oxidant system and oxidative stress, especially  during periods of higher pH, salinity, and temperature. These data also justify the use of P. globosa as a biomonitor of environmental conditions.

Reply: We are highly thankful to the reviewer for the encouraging comments. 

Overall, the manuscript, tables, and figures are clear and well-organized. The manuscript needs to be carefully edited, as there are some lengthy and awkward sentences and unclear wording in some sections. The results are very interesting and the manuscript should be accepted for publication.

Reply: Yes we agree that the ms had to be edited extensively. Therefore, we have modified almost the entire ms and all modifications are shown in red coloured text.